# Emergence of catalytic function in prebiotic information-coding polymers

Alexei V Tkachenko[1]*, Sergei Maslov[2,3,4]*

[1]Center for Functional Nanomaterials, Brookhaven National Laboratory, Upton, United States; [2]Department of Bioengineering, University of Illinois Urbana-Champaign, Urbana, United States; [3]Department of Physics, University of Illinois Urbana-Champaign, Urbana, United States; [4]Carl R Woese Institute for Genomic Biology, University of Illinois Urbana-Champaign, Urbana, United States

*For correspondence:
oleksiyt@bnl.gov (AVT);
maslov@illinois.edu (SM)

Competing interest: The authors declare that no competing interests exist.

**Abstract** Life as we know it relies on the interplay between catalytic activity and information processing carried out by biological polymers. Here we present a plausible pathway by which a pool of prebiotic information-coding oligomers could acquire an early catalytic function, namely sequence-specific cleavage activity. Starting with a system capable of non-enzymatic templated replication, we demonstrate that even non-catalyzed spontaneous cleavage would promote proliferation by generating short fragments that act as primers. Furthermore, we show that catalytic cleavage function can naturally emerge and proliferate in this system. Specifically, a cooperative catalytic network with four subpopulations of oligomers is selected by the evolution in competition with chains lacking catalytic activity. The cooperative system emerges through the functional differentiation of oligomers into catalysts and their substrates. The model is inspired by the structure of the hammerhead RNA enzyme as well as other DNA- and RNA-based enzymes with cleavage activity that readily emerge through natural or artificial selection. We identify the conditions necessary for the emergence of the cooperative catalytic network. In particular, we show that it requires the catalytic rate enhancement over the spontaneous cleavage rate to be at least $10^2$–$10^3$, a factor consistent with the existing experiments. The evolutionary pressure leads to a further increase in catalytic efficiency. The presented mechanism provides an escape route from a relatively simple pairwise replication of oligomers toward a more complex behavior involving catalytic function. This provides a bridge between the information-first origin of life scenarios and the paradigm of autocatalytic sets and hypercycles, albeit based on cleavage rather than synthesis of reactants.

## eLife assessment

This **valuable** study uses a model to determine when catalytic self-replication of polymers can emerge from a random pool of replicating polymers. The model accounts for the folding and function of polymers in addition to abstract evolutionary dynamics, providing **solid** evidence for the claims of the authors. The work will be of relevance to those interested in the origin of life, artificial cells, and evolutionary dynamics.

## Introduction

One of the most intriguing mysteries in science is the origin of life. Despite extensive research in this area, we are still far from understanding how life has emerged on Earth. One promising hypothesis is the RNA world theory (*Gilbert, 1986*; *Doudna and Szostak, 1989*; *Orgel, 2004*; *Lincoln and Joyce, 2009*; *Robertson and Joyce, 2012*; *Higgs and Lehman, 2015*), inspired by the discovery of ribozymes (*Kruger et al., 1982*), that is, RNA molecules capable of enzymatic activity. According to

this hypothesis, all processes in early life were carried out by the RNA, which was used both to store information and catalyze biochemical reactions. In particular, specific ribozymes could have catalyzed the self-replication of arbitrary RNA sequences, a function currently performed by specialized protein-based enzymes. However, based on the results of the existing experiments (*Bartel and Szostak, 1993*; *Horning and Joyce, 2016*), such a catalytic function requires rather long and carefully designed RNA sequences, which are highly unlikely to arise spontaneously. In contrast, one of the simplest catalytic activities of ribozymes is their ability to cleave an RNA sequence at a specific site. Indeed, such ribozymes independently evolved in multiple branches of life (*de la Peña and García-Robles, 2010*) and have been shown to emerge rapidly and repeatedly from artificial selection (*Williams et al., 1995*; *Salehi-Ashtiani and Szostak, 2001*). DNA molecules have also been shown to be capable of site-specific cleavage targeting either RNA (*Breaker and Joyce, 1994*) or DNA sequences (*Silverman, 2005*; *Chandra et al., 2009*).

In this article, we consider a population of information-carrying polymers capable of templated non-enzymatic replication (*Szostak, 2012*; *Kim et al., 2021*). This may have been the state of the proto-RNA world before the emergence of ribozymes. This could involve heteropolymers chemically distinct from present-day RNA (*Kim et al., 2021*) and/or inorganic catalysts such as mineral surfaces (*Ferris, 2005*; *Jerome et al., 2022*). We demonstrate that (i) even spontaneous cleavage promotes replication by generating short fragments used as primers for templated growth and (ii) catalytic cleavage activity naturally emerges in these populations and gets selected by the evolution (*Lukin, 2010*).

In a series of previous studies, we have shown that non-enzymatic templated replication can lead to the formation of longer chains (*Tkachenko and Maslov, 2015*) as well as to a reduction in sequence entropy (*Tkachenko and Maslov, 2018*). Such a reduction in entropy has subsequently been observed experimentally for templated ligation of DNA oligomers (*Kudella et al., 2021*). This selection in sequence space can be seen as a first step toward Darwinian evolution. However, this does not necessarily imply the emergence of a catalytic function. In this article, we build on these findings and further investigate the potential for the evolution of catalytic activity in the proto-RNA world.

## Model and results

In our model, we consider the population dynamics of a pool of heteropolymers analogous to the familiar nucleic acids (RNA or DNA) but capable of enzyme-free templated polymerization. The basic processes in this scenario are similar to those in polymerase chain reaction (PCR), where the system is driven out of equilibrium by cyclic changes in the environment (e.g., temperature), which we refer to as 'night' and 'day' phases. During the night phase, heteropolymers hybridize with each other following Watson–Crick-like complementarity rules. If the terminus of one chain hybridizes with the middle section of another chain, the former can be gradually elongated by the virtue of non-enzymatic templated polymerization (*Sulston et al., 1968*; *Weimann et al., 1968*; *Lohrmann et al., 1980*; *Duzdevich et al., 2020*). We will refer to this type of hybridization as 'end-to-middle', the former chain as the 'primer' and the latter chain as the 'template'. During the day phase, the hybridized structures melt and all the heteropolymers separate from each other. During the next night, they hybridize with new partners, providing them with the opportunity to elongate further. Unlike the classical PCR process, we assume that the polymerization in our system occurs without any assistance from enzymes and may proceed in either direction along the chain. Equivalently, instead of polymerization, the elongation could rely on ligation with very short chain segments. It is important to note that in the context of RNA, such bidirectional elongation requires chemical activation of the phosphate group at the 5′ end of the primer to provide free energy for the newly formed covalent bond. Like the polymerization process itself, achieving this without enzymes is biochemically challenging. One might speculate that prebiotic evolution relied on inorganic catalysis, such as on mineral surfaces, or involved polymers other than today's RNA.

The elongation of primers naturally leads to the copying of information from the template's sequence. The obvious limiting factor for this process is the availability of primers and the likelihood of end-to-middle hybridization resulting in elongation. The key observation behind our model is that the breakup (cleavage) of a chain creates a new pair of potential primers. Each of them could be elongated during subsequent nights. Thus, somewhat counterintuitively, breakup of chains results in their proliferation.

Our previous theoretical (*Tkachenko and Maslov, 2015*; *Tkachenko and Maslov, 2018*) and experimental (*Kudella et al., 2021*) results demonstrated that templated-assisted replication of heteropolymers has a generic tendency to substantially reduce their sequence entropy. Such reduction has important consequences in the context of the current work: it significantly increases the likelihood of end-to-middle hybridization of chains during the night phase. That in turn creates an evolutionary pressure to further decrease sequence entropy. A detailed study of this fascinating mechanism falls beyond the scope of the current study. However, below we will assume the logical end of this dynamics where the pool of sequences is composed of fragments of one or several nearly non-overlapping master sequences and their complementaries. Note that for any two overlapping chains, such that the sequence of the first one is a fragment of the master sequence while the other is a fragment of the complementary master sequence, the end-to-middle binding is essentially guaranteed. The exception to this rule is when both chains terminate at the same points so that they are exact complements of each other.

## Random cleavage model

Based on the argument presented above, we focus on the case of a system populated with chains that are fragments of a single master sequence or its complement. We denote the total concentration of fragments of the master sequence as $c(t)$, while the concentration of all fragments of the complementary sequence as $\bar{c}(t)$. Our system operates in a chemostat, that is, a reservoir constantly supplied with fresh monomers at the concentration $m_0$ and diluted at the rate $\delta$. Let $M(t)$ (respectively $\bar{M}(t)$) be the concentration of all monomers incorporated into chains of the subpopulation $c(t)$ (respectively $\bar{c}(t)$). The concentration of free monomers not incorporated into any chains is given by:

$$m = m_0 - M - \bar{M} \tag{1}$$

It is convenient to introduce a minimal length $l_0$ of a chain that would hybridize with its complementary partner during the night phase, and use it as the unit of chain length, instead of a single monomer. In effect, this leads to the renormalization of all monomer concentrations as $m = [m]/l_0$, where $[m]$ is the conventional molarity. $M$, $\bar{M}$, and $m_0$ are similarly renormalized, while the polymer concentration remains unmodified: $c = [c]$. In what follows, we renormalize all lengths and concentrations so that $l_0 = 1$.

We assume that nights are sufficiently long and that the hybridization rate is sufficiently fast so that early on during each night phase most chains find partners with a complementary overlap. That assumes that the total concentration of all master sequence fragments, $c$, is lower than the total concentration of all fragments in the complementary subpopulation, $\bar{c}$. If this is not the case, that is, if $c > \bar{c}$, only a fraction $\bar{c}/c$ of all fragments of the master sequence find a partner, while the rest of the $c$-subpopulation remains unpaired and thus does not elongate. For simplicity, our model neglects the possibility of the formation of hybridized complexes involving more than two chains.

It is well known that self-replication based on template-based polymerization or ligation is vulnerable to product inhibition, that is, re-hybridization of the products intended to act as templates (*Szostak, 2012*; *Tupper and Higgs, 2021*). Indeed, since full-length templates and their complements would bind more strongly to each other than any shorter fragment, the primers would typically be displaced from the templates by longer chains, leading to effective 'template' poisoning. In our model, however, the master sequence fragments act as both primers and templates. Of two oligomers in a hybridized pair, the one that extends further in a particular direction acts as a template for the growth of the other. At the same time, the partner may also act as a template for the growth of the first oligomer in the other direction. In either case, there will typically be two growing ends per duplex. Full inhibition occurs only when two hybridized chains terminate at exactly the same point in both directions. Since this is a rather rare event, we neglect its effect.

As discussed earlier, two hybridized chains whose sequences are fragments of the master sequence and its complementary would typically have two ends that undergo templated growth at a certain rate proportional to monomer concentration $m$ (see *Figure 1*). The exception to this rule is when these two hybridized chains terminate at exactly the same point. In our model, the average primer elongation rate is $r \cdot m(t)$. The rate parameter $r$ accounts for the finite probability of primer binding to a template during the night phase as well as for a finite night-to-day ratio. Note that the value of $r$ does not change after our renormalization $l_0 \to 1$. The $M(t)$ dynamics is thus given by

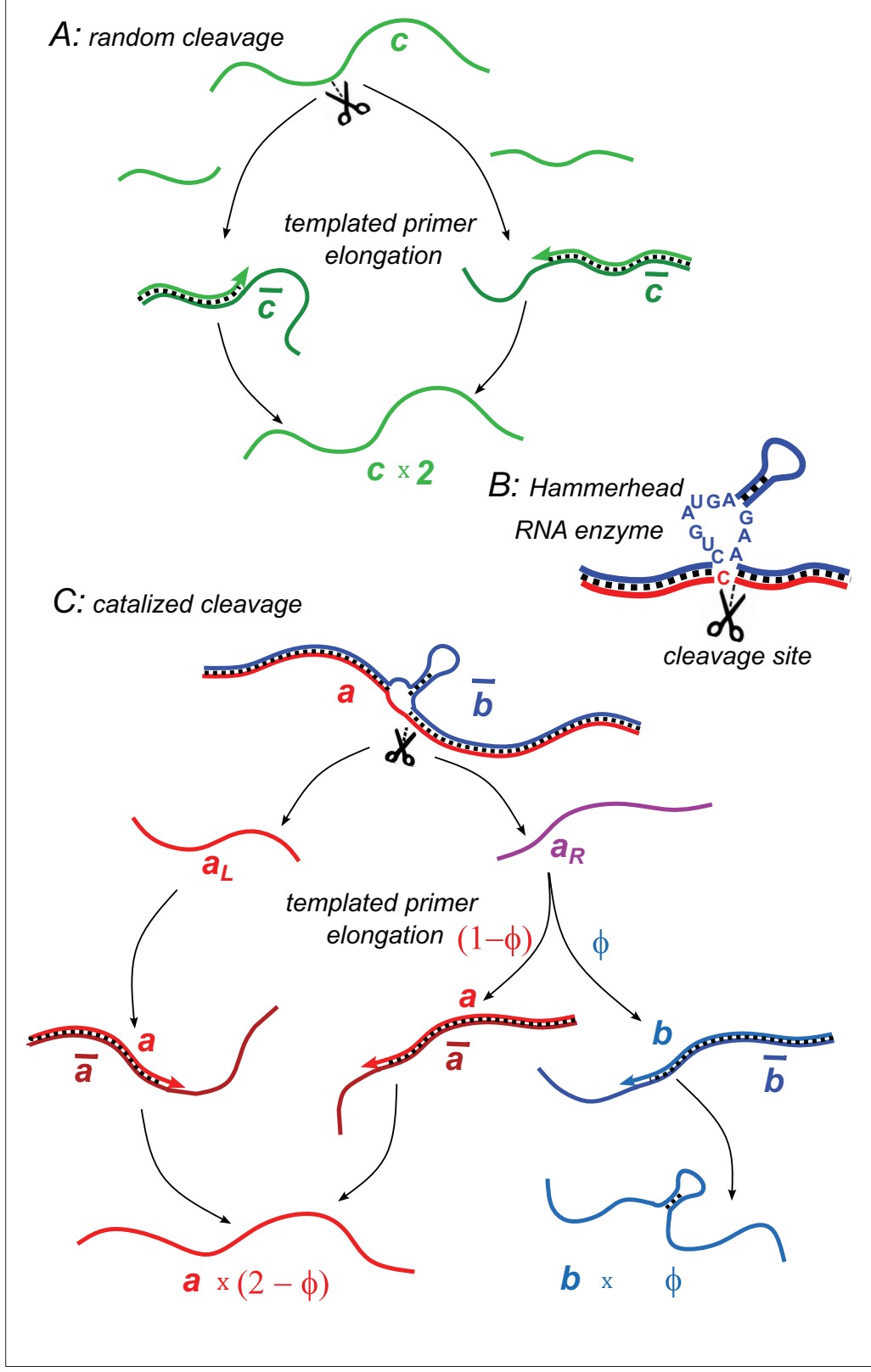

**Figure 1.** Illustration of two model variants. (**A**) Random cleavage model. A random break in a chain of type $c$ generates two primer fragments, which are elongated to give rise to two chains of type $c$. Elongation requires a complementary template of type $\bar{c}$. (**B**) An example of catalyzed cleavage given by hammerhead ribozyme (**Pley et al., 1994**). Note that the right cleavage fragment is perfectly complementary to the blue sequence,

*Figure 1 continued on next page*

*Figure 1 continued*

while the left one contains an extra non-complementary base $C$. (**C**) Catalyzed cleavage model. A cleavage of the red chain $a$ catalyzed by the blue chain $\bar{b}$ gives rise to two primers $a_L$ (red) and $a_R$ (purple). Because of an extra non-complementary base (see **B**), the $a_L$ primer can only elongate to $a$, while the $a_R$ primer – to either $a$ or $b$ depending on its first hybridization partner. Similar processes involving complementary chains $\bar{a}$ and $\bar{b}$ (not shown) result in the replication of templates.

$$\frac{dM}{dt} = rm \cdot \min(c, \bar{c}) - \delta \cdot M \tag{2}$$

The equation for the complementary subpopulation is obtained by replacing $M$ with $\bar{M}$. Note that $m$ includes only monomers, and therefore is not subdivided into two complementary subpopulations.

We assume that the breakup of chains is completely random and happens at a constant rate $\beta_0$ at any internal bond along the chain. The concentration of these breakable bonds is given by $M$. Because of our choice of the unit length $l_0$, $\beta_0 = l_0 \beta_0'$, where $\beta_0'$ is the cleavage rate of a single bond. Since each fragmentation of a chain creates one new primer, the equation governing the overall concentration of chain fragments in the subpopulation $c$ is given by

$$\frac{dc}{dt} = \beta_0 M - \delta c \tag{3}$$

Once again, the equation for the complementary subpopulation is obtained by replacing $c$ with $\bar{c}$ and $M$ with $\bar{M}$.

Combining *Equations 2 and 3*, we observe that the steady state is a symmetric mixture $c = \bar{c}$. The average length (in units of $l_0$) of all chain fragments in the subpopulation $c$ is given by

$$\langle L \rangle = \frac{M}{c} = \frac{\delta}{\beta_0} = \frac{rm^*}{\delta}. \tag{4}$$

This in turn determines the steady-state concentrations:

$$m^* = \frac{\delta^2}{\beta_0 r} \tag{5}$$

$$c = \bar{c} = \frac{1}{2}\left(\frac{\beta_0}{\delta}m_0 - \frac{\delta}{r}\right) \tag{6}$$

To obtain *Equation 6*, we combined *Equations 1 and 4*. Note that the mutually templating chains survive only when the concentration of free monomers supplied to the system $m_0$ exceeds $m^*$.

## Model with catalyzed cleavage

In the model considered above, the random breakage of chains led to their proliferation. It is therefore reasonable to expect that the ability of a heteropolymer to catalyze cleavage would be selected by the evolution. Incidentally, some of the simplest known RNA-based enzymes (ribozymes) have exactly this function (*Kruger et al., 1982*; *Prody et al., 1986*; *Hutchins et al., 1986*; *Scott et al., 1995*; *Williams et al., 1995*; *Salehi-Ashtiani and Szostak, 2001*; *de la Peña and García-Robles, 2010*; *Scott et al., 2013*).

Here we consider a simple model in which a heteropolymer capable of catalyzing cleavage spontaneously emerges from a pool of mutually templating chains. Our model is inspired by the real-world examples of naturally occurring hammerhead ribozyme (*Prody et al., 1986*; *Hutchins et al., 1986*), as well as artificially selected DNA-cleaving DNA enzymes (*Chandra et al., 2009*). The minimal structure of the hammerhead ribozyme (*Pley et al., 1994*; *Scott et al., 1995*; *Scott et al., 2013*) consists of a core region of 15 (mostly) invariant nucleotides flanked by three helical stems formed by mutually complementary RNA sequences. The cleavage happens at a specific site of this structure, located immediately adjacent to one of these stems. While a classical hammerhead ribozyme consists of a single RNA chain capable of self-cleavage, the same structure could be assembled from two chains, one (labeled $b$ in *Figure 1*) containing a hairpin and capable of hybridizing with and subsequently cleaving the other chain (labeled $a$ in *Figure 1*). Furthermore, such two-chain structuresare realized in certain DNA-cleaving DNA enzymes (*Chandra et al., 2009*).

We consider a scenario in which a master sequence spontaneously emerges from a random pool and subsequently diverges into two closely related subpopulations $a$ and $b$ and their respective complementarities $\bar{a}$ and $\bar{b}$. The sequences of $a$ and $b$ are mostly identical except for a short insert in the chains $b$ and $\bar{b}$, rendering them catalytically active. That is to say, when $\bar{b}$ is bound to a chain from the subpopulation $a$, it induces a cleavage at a specific site of that chain. Inspired by the hammerhead ribozyme, we assume that the cleavage site in $a$ is immediately adjacent to the start of the catalytic insert in $b$ (see *Figure 1*).

We further assume that $b$-chains are capable of cleaving $\bar{a}$-chains at the same site. This symmetry most likely does not apply to highly optimized hammerhead ribozymes, but it is reasonable in enzymes with only a modest level of catalytic efficiency. Let the cleavage within $\bar{b}$-$a$ or $b$-$\bar{a}$ duplexes occur at rates $\beta$ and $\bar{\beta}$, respectively.

The cleavage of a chain of type $a$ by $\bar{b}$ produces two pieces, right and left. The concentrations of these fragments are referred to as $a_R$ and $a_L$, respectively. Inspired by the example of the hammerhead ribozyme (illustrated in *Figure 1B*), we assume that one of them ($a_L$) can serve as a primer only for $a$. Conversely, the other fragment ($a_R$) can serve as a primer for either $a$ or $b$ depending on the first templating chain with which it will hybridize (respectively $\bar{a}$ or $\bar{b}$). Assuming a random first encounter, the probability for $a_R$ to serve as a primer to $b$ is thus given by $\phi = \bar{b}/(\bar{a} + \bar{b}) < 1$, while the probability of it to grow into $a$ is $1 - \phi = \bar{a}/(\bar{a} + \bar{b})$. Similarly, conversion probabilities in the complementary subpopulation $\bar{a}_R$ are determined by $\bar{\phi} = b/(a + b)$.

At this point, the issue of template poisoning due to product rehybridization should be revisited. In the context of random cleavage, we have argued that it is avoided due to the low probability of binding of two fragments terminating at exactly the same sites. However, catalyzed cleavage produces primers $a_R/\bar{a}_R$ and $a_L/\bar{a}_L$ that terminate in the same region, and therefore they would typically be displaced from the respective templates in favor of the formation of $a - \bar{a}$ and $b - \bar{b}$ duplexes. In Appendix 1, we analyze the binding kinetics during the night phase and come to an important conclusion. If the concentration of primers (such as $a_R$) is small compared to that of templates (such as $\bar{a}$), despite the strand displacement, a finite and substantial fraction of primers will remain hybridized to their respective templates for a significant time. More specifically, they would remain hybridized until the concentration of free templates drops to the level of the concentration of primers. This indicates that template poisoning has only a moderate effect if the night phase is not too long.

As in the random cleavage model, the concentration of hybridized duplexes is given by the smaller of two concentrations $a + b$ and $\bar{a} + \bar{b}$. This is captured by the factor $\chi = \min(a + b, \bar{a} + \bar{b})/(a + b) \leq 1$ in the elongation rate of a primer: $rm\chi$. Here, as before, this rate is measured in $l_0$ bases per unit time, and $m$ is the free monomer concentration.

We observe that in order for a cleavage fragment to work as a primer, it needs to exceed the minimal primer length $l_0$. Therefore, a newly formed cleavage product $a_L$ needs to grow by at least $l_0$ bases before it can be considered a part of the $a$ subpopulation. Indeed, if it has not grown by that length, another catalyzed cleavage at the same site would not increase the number of primers in the system. Therefore, the rate at which chains in the subpopulation $a_L$ are converted to $a$ is given by $rm\chi$. The rate of conversion of $a_R$ to $a$ is similar, up to the factor $1 - \phi$ discussed above: $rm\chi(1 - \phi)$. If a segment $a_R$ was first hybridized to $\bar{b}$, it will eventually grow to be a part of the subpopulation $b$. However, in order to become functional, this chain has to grow at least by the length of the catalytic insert, which is distinct from $l_0$. Furthermore, the rate of elongation of $b$ is slowed down by the presence of a hairpin in the catalytic domain of the $\bar{b}$ structure. Both effects can be captured by a factor $\lambda$ in the conversion rate from $a_R$ to $b$ given by $rm\chi\phi/\lambda$, relative to that of $a_R$ to $a$ given by $rm\chi(1 - \phi)$.

Altogether, the dynamics of our model is described by the following equations:

$$\dot{c} = \beta_0 M + \beta\phi a - \delta c \tag{7}$$

$$\dot{b} = \beta_0 M\bar{\phi} + \frac{rm\chi\phi}{\lambda}a_R - \delta b \tag{8}$$

$$\dot{a}_L = \beta\phi a - rm\chi a_L - \delta a_L \tag{9}$$

$$\dot{a}_R = \beta\phi a - rm\chi(1 - \phi)a_R - \frac{rm\chi\phi}{\lambda}a_R - \delta a_R \tag{10}$$

Here, $c \equiv a + b + a_l + a_R$. Note that *Equation 7* is obtained by first writing the kinetic equation for $a$ and then adding it up to the sum of *Equations 8–10*. The first terms in the r.h.s. of *Equations 7 and 8*

represent random non-catalyzed cleavage that occurs at rate $\beta_0$ at any location of any chain (compare to *Equation 3*). Similarly to *Equation 2*, the dynamics of the number density $M$ of monomers incorporated into chains $a$, $a_L$, $a_R$, and $b$ is described by

$$\dot{M} = rm\chi c - \delta M. \tag{11}$$

This in turn determines the concentration of free monomers remaining in the solution as given by *Equation 1*.

Additional equations are obtained by replacing variables $a$, $a_L$, $a_R$, $b$, and $M$ with their complementary counterparts: $\bar{a}$, $\bar{a}_L$, $\bar{a}_R$, $\bar{b}$, and $\bar{M}$, respectively. In addition, β, φ, and χ should be replaced with $\bar{\beta}$, $\bar{\phi}$, and $\bar{\chi} = \min(a + b, \bar{a} + \bar{b})/(\bar{a} + \bar{b})$, respectively.

## System dynamics

A crucial feature of this multicomponent system is that catalytic cleavage depends on the cooperativity between all four subpopulations $a$, $\bar{a}$, $b$, and $\bar{b}$. To understand when such a cooperative steady state exists, we have numerically solved *Equations 7–11* for $\beta_0 = 0.015$, $\lambda = 2$, $\delta = 1$ and different values of the catalytic cleavage rate $\beta$. *Figure 2A–C* show the dynamic trajectories of our system in $a$–$b$ space for a wide range of initial conditions. For $\beta$ too small (e.g., $\beta = 6$ in *Figure 2A*) or too large (e.g., $\beta = 18$ in *Figure 2C*), the only steady-state solutions correspond to the survival of either the $a/\bar{a}$ or $b/\bar{b}$ subpopulation. In these non-cooperative fixed points, marked with red and blue stars in *Figure 2*, one set of chains drives the other to extinction. Thus, they propagate by random rather than catalytic cleavage. The concentrations of monomers and a complementary pair of surviving chains at a non-cooperative fixed point are given by *Equations 5 and 6*, respectively.

For an intermediate value of β (e.g., for $\beta = 10$ shown in *Figure 2B*), we observe the emergence of a new cooperative fixed point (marked by the green star), where all four subpopulations survive at concentrations $a = \bar{a} > 0$ and $b = \bar{b} > 0$. This fixed point is mainly maintained by catalytic cleavage.

The phase portrait of our system shown in *Figure 2B* suggests a plausible scenario for the emergence and subsequent evolution of this cooperative system. This evolutionary scenario starts with $b/\bar{b}$ subpopulations existing alone. Eventually, copying errors might result in the emergence of a small subpopulation of $a$ or $\bar{a}$ at a concentration $a \ll b$. If this concentration is greater than a certain threshold separating blue and green trajectories in *Figure 2B*, the dynamics of the system would drive it to the cooperative fixed point (green star in *Figure 2B*). For specific parameters used to generate *Figure 2B*, the minimal ratio between concentrations $a$ and $b$ is roughly 0.01.

## Properties of the cooperative steady state

To better understand the conditions for the existence of the cooperative regime, we analytically derived the steady-state solutions of *Equations 7–11*. The key result is the relationship between the steady-state monomer concentration, $m^*_{coop}$, and the catalytic cleavage rate β for given values of $\beta_0$, $\lambda$, δ, and $r$ derived in Appendix 1. *Figure 3A* shows this dependence for $\lambda = 2$, and different values of $\beta_0/\delta$ alongside with data points (open circles) obtained by direct numerical solution of dynamical (*Equations 7–11*).

The stable fixed point corresponds to the monotonically increasing branch of the graph $m^*_{coop}$ vs. β (solid lines in *Figure 3A*), while two decreasing branches (dashed lines in *Figure 3A*) correspond to two dynamically unstable saddle points separating different steady-state solutions (see *Figure 2B*).

Note that the stability of our cooperative fixed point is a non-trivial result. For example, in a related model by *Kamimura et al., 2019*, the fixed point corresponding to a viable composite replicase is dynamically unstable and requires additional stabilization, for example, by cell-like compartments.

Increasing the parameter $\beta_0/\delta$, for example, by making the dilution rate δ smaller, makes the range of β for which the cooperative solution exists progressively smaller until it altogether disappears above $\beta_0/\delta \approx 0.057$ (for $\lambda = 2$).

*Figure 3B* shows the ranges of $\beta/\delta$ and $\beta_0/\delta$ for which the cooperative solution exists (as before, for $\lambda = 2$). Solid lines of different colors correspond to three values of $\beta_0$ used in *Figure 3A*.

While the full set of our analytical results described in Appendix 1 is rather convoluted, here we present a simplified expression for the range of values of $\beta/\delta$, where the cooperative solution exists, and the corresponding range of $m^*_{coop}$:

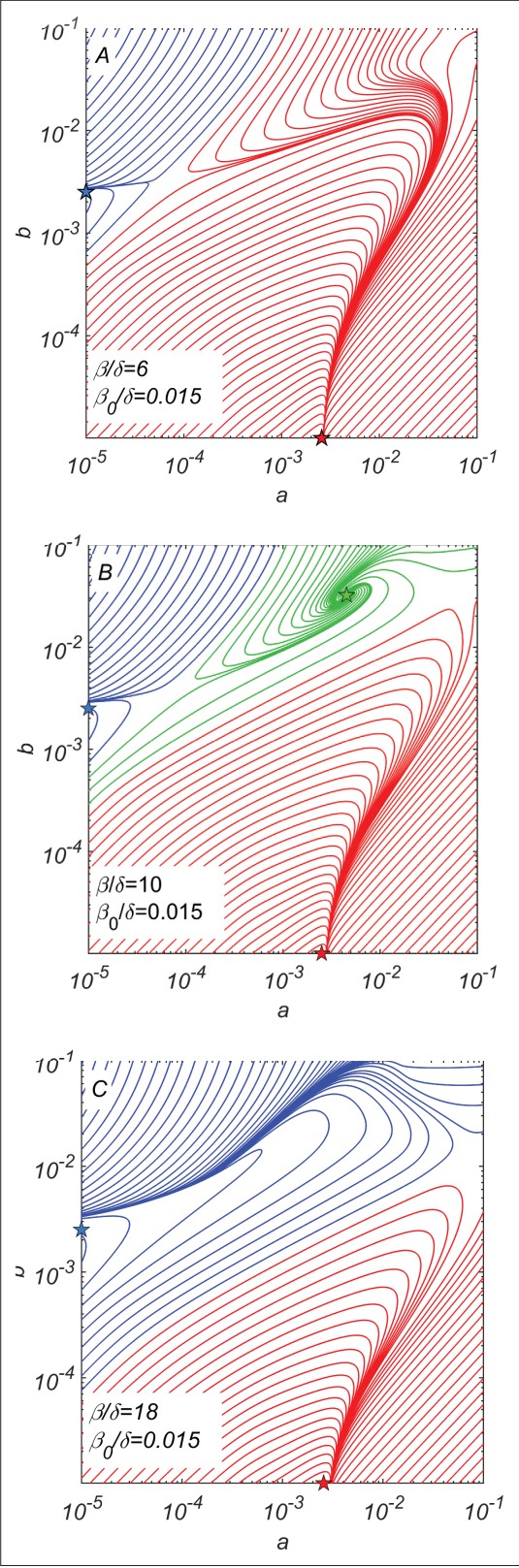

**Figure 2.** Dynamical phase portraits for different catalytic cleavage rates β. (**A**) The phase portrait for a small catalytic cleavage rate $\beta = 6$ has two non-cooperative steady-state solutions marked with red and blue stars corresponding to pure $a/\bar{a}$ and pure $b/\bar{b}$ subpopulations, respectively. These solutions are maintained by random rather than catalytic cleavage. (**B**) The phase portrait for intermediate catalytic cleavage rate $\beta = 10$ in addition to

*Figure 2 continued on next page*

*Figure 2 continued*

two non-cooperative steady states marked with red and blue stars has a cooperative steady state marked with the green star in which all four subpopulations coexist. One can reach this state, for example, starting from the non-cooperative $b/\bar{b}$ steady state (the blue star) and adding a relatively small subpopulation of $a/\bar{a} > 2e - 5$ crossing the saddle point separating blue and green trajectories. (**C**) The phase portrait for a large catalytic cleavage rate $\beta = 18$ again has only two non-cooperative cleavage steady states marked with red and blue stars. All three panels were obtained by numerically solving dynamical *Equations 7–11* with random cleavage rate $\beta_0 = 0.015$, elongation asymmetry factor $\lambda = 2$, and dilution factor $\delta = 1$.

$$4\lambda < \frac{\beta}{\delta} < \frac{(\lambda - 1)\delta}{4\beta_0} \tag{12}$$

$$\frac{2\lambda\delta}{(\lambda - 1)\beta_0} < \frac{m^*_{\text{coop}}}{m^*} < \frac{1}{2} \tag{13}$$

These conditions were derived in the limit $\beta_0/\delta \ll 1/\lambda$ (see Appendix 1 for details).

## Evolutionary dynamics

The competition for monomers is the main mechanism of natural selection operating in our system. The steady state with the lowest level of monomer concentration is favored by evolution since the competing states would not be able to proliferate at that level of monomers. *Equation 13* implies that the monomer concentration $m^*_{\text{coop}}$ in the cooperative solution is always less than a half of its value $m^*$ in the absence of the catalytic cleavage. Thus, once the cooperative state emerges, it drives out all chain sequences that still rely on non-catalytic cleavage for replication. The continuously increasing 'fitness' of the system can be quantified, for example, by the ratio $m^*/m^*_{\text{coop}}$. Note that the non-cooperative solution has a fitness of 1, while the cooperative solution has a fitness higher than 2. Note that this fitness is defined at the level of the ecosystem comprising all the sequences in the chemostat and may not necessarily be attributable to individual members of that population. Over time, similar to microbial ecosystems, this population changes according to the laws of competitive exclusion (*Gause, 1934*; *Tilman, 1982*).

The fitness landscape of our system shown in *Figure 4* depends on two parameters of the catalytic cleavage: β and $\lambda$. Its preeminent feature is a relatively narrow fitness ridge (orange and red color in *Figure 4*). For a fixed value of $\lambda$, this ridge corresponds to a sharp fitness maximum located at the lowest possible catalytic cleavage rate β for which the cooperative state still exists (see the red region in *Figure 4B*). In other words, for a given $\lambda$, the selective pressure would drive β down to the lower boundary separating cooperative and non-cooperative regions.

However, $\lambda$ and β are expected to co-evolve together. Indeed, $\lambda > 1$ quantifies the ratio by which the structural properties of the type-*b* chain (its excess length, the hairpin unzipping free energy, etc.) slow down its replication compared to that of the type-*a* chain. Thus, it is reasonable to assume that $\lambda$ could be easily modified in the course of the evolution. As a consequence of the selective pressure, both $\lambda$ and β are expected to increase in the course of the evolution driving the system up the ridge in *Figure 4*. This intuition is confirmed by a direct numerical simulation in which we model the evolution as a Monte Carlo process with fitness playing the role of negative energy Parameters $\lambda$ and β were allowed to vary randomly and independently of each other. A sample evolutionary trajectory is shown as a dashed line in *Figure 4*.

One can imagine several possible pathways leading to this self-sustaining cooperative system. (i) A pair of mutually complementary non-catalytic chains (*a* and $\bar{a}$ in our notation) gains function due to a copying error, giving rise to a small subpopulation of 'sister' chains (*b* and $\bar{b}$ in our notation) with nascent cleavage activity directed toward $\bar{a}$ and *a*, respectively. (ii) A pair of catalytically active chains *b* and $\bar{b}$ emerges first, subsequently losing the catalytic inserts due to a copying error, thus giving rise to a small subpopulation of substrate chains *a* and $\bar{a}$. According to the dynamic phase portrait of our system illustrated in *Figure 2B*, the second pathway is more plausible than the first one. Indeed, at least for the parameters used in *Figure 2B*, the conversion of only a few percent of *b* chains to *a* brings the systems into the basin of attraction of the cooperative fixed point marked green in the figure. The other scenario is less likely since it requires a much higher ratio of emergent over ancestral population sizes.

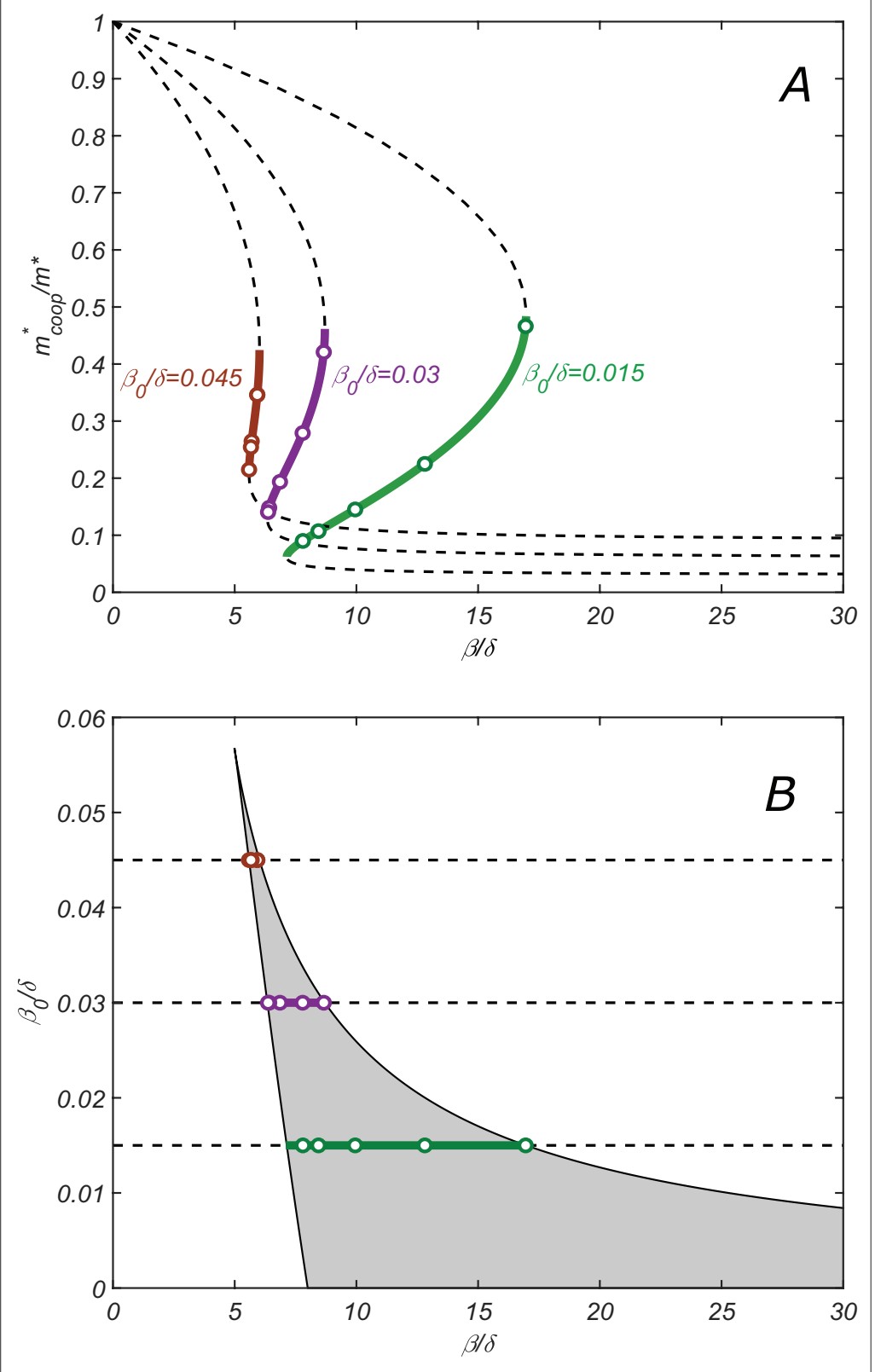

**Figure 3.** Properties of the cooperative state. (**A**) The relationship between parameters of the cooperative state. $m^*_{coop}/m^*$ plotted vs. $\beta$ for $\lambda = 2$, $\delta = 1$ and increasing values of $\beta_0$: 0.015 (green), 0.003 (purple), and 0.0045 (red). Lines are given by the parametric equation describing the state and derived in the SI Appendix (*Equation S14*), while open circles are obtained by direct numerical solution of dynamical *equations (7–11)*. Monotonically

*Figure 3 continued on next page*

*Figure 3 continued*

increasing branches (solid lines) correspond to the stable cooperative fixed point, while the decreasing branches (dashed lines) – to the dynamically unstable saddle points separating different steady-state solutions in *Figure 2B*. (**B**) Phase diagram of the cooperative state. The shaded region marks the values of $\beta/\delta$ and $\beta_0/\delta$ for which the cooperative solution exists. Green, purple, and red lines show the ranges of $\beta$ for which the cooperative solution exists for the corresponding value of $\beta_0$ shown in (**A**). Increasing the parameter $\beta_0$ makes the range of $\beta$ for which the cooperative solution exists progressively smaller until it altogether disappears above. $\beta_0/\delta \approx 0.057$.

Our results indicate that the cooperative steady state emerges when the catalytic rate enhancement over the spontaneous cleavage rate is at least $10^2$–$10^3$. This is a relatively modest gain compared to the $10^9$ enhancement reported for a highly optimized hammerhead ribozyme *Scott et al., 2013*. However, it is comparable to the rate enhancement observed after only five rounds of in vitro selection from an unbiased sample of random RNA sequences (*Salehi-Ashtiani and Szostak, 2001*). In the course of subsequent evolution, two main parameters of our model β and $\lambda$ are expected to increase in tandem. The first parameter, β, is the catalytic cleavage rate, which, as we know, can increase over multiple orders of magnitude (*Salehi-Ashtiani and Szostak, 2001*). The second parameter, $\lambda$, quantifies the relative delay in the elongation of $b$-types chains compared to $a$-type chains. It may be caused at least in part by the difficulty of unzipping the hairpin, thus $\lambda$ can be dramatically increased by making the hairpin stem longer. One should note that our model requires the catalytic activity of both $b$ chains and their complementary partner, $\bar{b}$ chains. Thus, sequences are expected to evolve to simultaneously optimize these two catalytic rates. The need for this compromise would likely prevent the catalysts from reaching the maximum efficiency observed, for example, in fully optimized ribozymes (*Scott et al., 2013*). The drive to further optimize cleavage activity might trigger a transition to more complex catalytic networks, for example, to an increase in the number of chains involved.

## Discussion

The proposed scenario is certainly not the only plausible pathway that could lead to the emergence of functional heteropolymers in the prebiotic world. To illustrate some of the unique features of the presented model, it is useful to place it in the context of other recent proposals. One of the most intriguing possibilities is the virtual circular genome (VCG) model recently proposed in *Zhou et al., 2021*. It is based on the observation that a relatively long ancestral genome can be stored as a collection of short overlapping RNA fragments of a circular master sequence or its complement. The model assumes unidirectional non-enzymatic replication of these fragments. On the one hand, it explains how a collection of relatively short RNA fragments (10–12 nucleotides each) could store a large amount of genetic information. On the other hand, computer simulations indicate that the VCG is susceptible to so-called sequence scrambling, where the appearance of repeats in the master sequence results in the loss of integrity of an entire circular genome (*Chamanian and Higgs, 2022*). The proliferation of the VCG is quite sensitive to sequence scrambling since the model assumes unidirectional polymerization, so that in order to return to copying a particular segment, one must copy the entire virtual circle.

Our model, while sharing some elements with the VCG model, differs in two important aspects: (i) it assumes that the non-enzymatic templated polymerization is bidirectional, and (ii) the functional activity of the heteropolymer is localized in a relatively short sequence region that is catalytically cleaved and thus replicated first. This implies that possible scrambling outside of the narrow functional region does not affect the viability of our autocatalytic system. Indeed, assuming the minimal hybridization length $l_0 = 6$ and random statistics of the master sequence, one gets the scrambling-free length $\sqrt{2 \cdot 4^l_0} + l_0 \simeq 100$. This is an order of magnitude larger than both $l_0$ and the length of the core region of the hammerhead ribozyme.

It may be possible to incorporate the selection mechanism proposed in this article into the VCG model. Such a hybrid approach would avoid the need for the biochemically problematic bidirectional growth, while explaining the emergence of early catalytic activity unaffected by sequence scrambling. This in turn could pave the way for the emergence of the rolling cycle, a model proposed as an alternative to VCG. (*Tupper and Higgs, 2021*; *Rivera-Madrinan et al., 2022*). The rolling cycle relies on strand displacement rather than cyclic melting of hybridized duplexes and requires a more

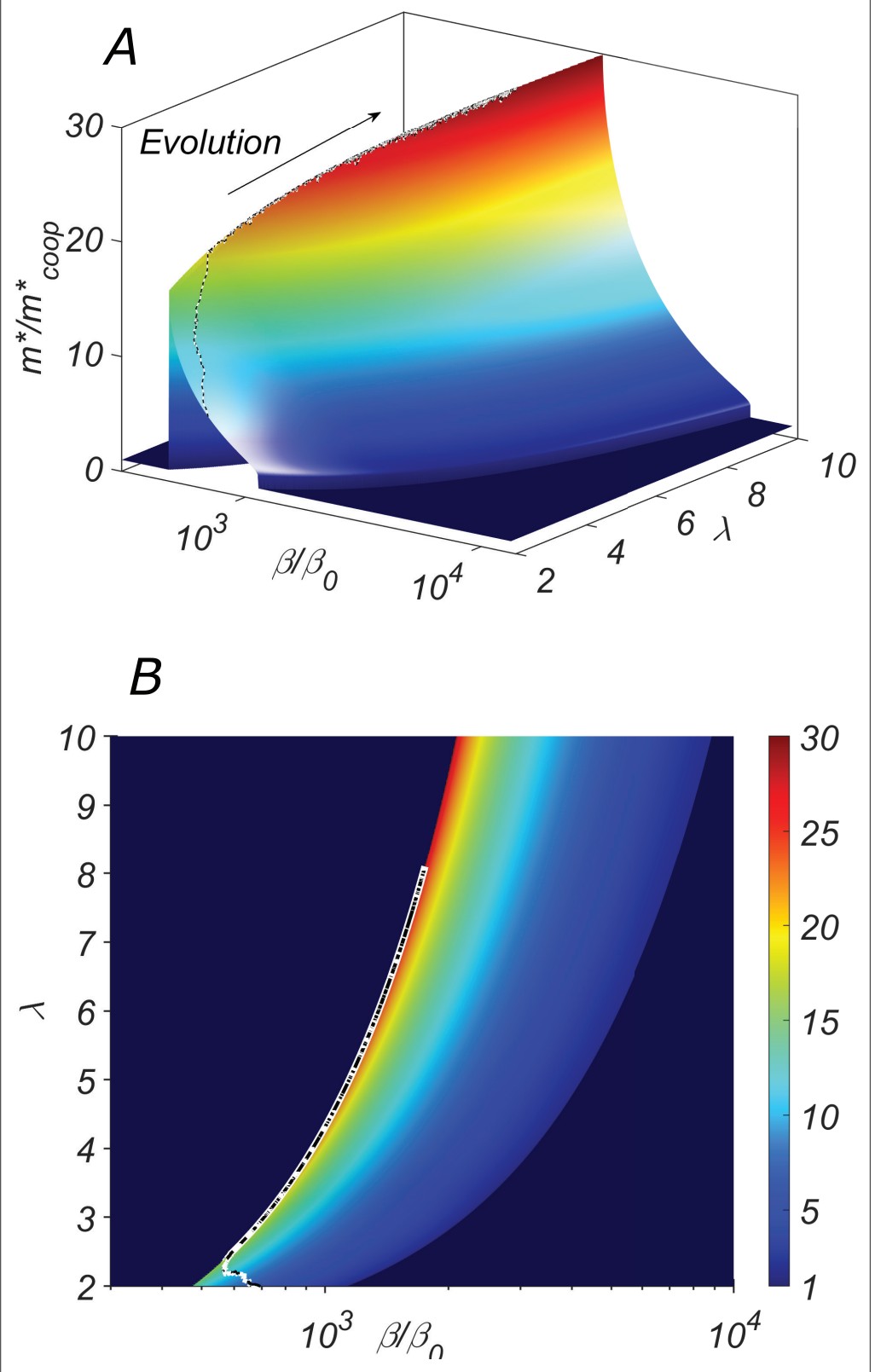

**Figure 4.** The fitness landscape of our system. (**A**) the three-dimensional plot and (**B**) the heatmap of the fitness $m^*/m^*_{\text{coop}}$ of the cooperative state in our system as a function of the catalytic cleavage rate enhancement $\beta/\beta_0$ and elongation asymmetry $\lambda$. The co-evolution of $\lambda$ and $\beta$ would increase together. A typical evolutionary trajectory in which small changes in these parameters are independent of each other is shown as a dashed line.

sophisticated setup, including a pre-selected sequence pool and the availability of cleavage enzymes. Our model provides a plausible pathway for how this catalytic function could have evolved.

Our model aims to describe the early stages of the evolution of life on Earth based on non-enzymatic polymerization. While it may seem challenging to test it for conditions relevant to the origin of life, our main conclusions can still be verified experimentally. RNA or DNA can be used as model polymers in such experiments, as both have demonstrated catalytic abilities in cleavage reactions (*de la Peña and García-Robles, 2010*; *Williams et al., 1995*; *Salehi-Ashtiani and Szostak, 2001*; *Breaker and Joyce, 1994*; *Silverman, 2005*; *Chandra et al., 2009*). To simulate primordial polymerization driven by day/night cycling, the experiment would have to rely on enzymatic polymerization or ligation as used, for example, in PCR (*Mullis and Faloona, 1987*; *Saiki et al., 1988*) or ligase chain reaction (LCR) (*Barany, 1991*). However, it is important to note that our model assumes polymerization in both the 5′-to-3′ and 3′-to-5′ directions, unlike traditional PCR, which only adds new nucleotides in the 5′-to-3′ direction. This problem of bidirectional polymerization was solved by evolution using Okazaki fragments (*Okazaki et al., 1968*). Inspired by this discontinuous synthesis of the lagging strand of DNA, we propose a possible experimental implementation of our system based on ligation rather than polymerization enzymes. In this scenario, the system would be supplied with ultrashort random DNA segments. These segments, which are much shorter than the minimal primer length ($l_0$), would play the role of 'monomers' and bidirectional primer extension would occur through a sequence of ligation steps connecting adjacent ultrashort segments to each other. Another important consideration for experimental implementation is the need to activate the nucleotides to provide free energy for polymerization. Thus, both the short fragments supplied to the system and the new primers formed by cleavage must be chemically activated.

## Conclusions

Our results suggest a plausible pathway by which a pool of information-carrying polymers could acquire the catalytic function, thereby bringing this system closer to the onset of the RNA world. We start with a pool of polymers capable of non-enzymatic templated polymerization and subjected to cyclic change of conditions (day/night cycles). The replication in our system is carried out during the night phase of the cycle via the elongation of primers hybridized with their complementary templates. We first observe that any cleavage of chains generates new primers and thus promotes their replication. The mutual replication of complementary chains is sustainable as long as cleavage and elongation rates are large compared to the dilution rate: $\beta_0 m_0 r > \delta^2$. This suggests that a faster, catalyzed cleavage would be selected by the evolution. Furthermore, DNA or RNA sequences capable of catalyzing site-specific cleavage are known to be relatively simple and readily arise via either natural or artificial selection (*de la Peña and García-Robles, 2010*; *Williams et al., 1995*; *Salehi-Ashtiani and Szostak, 2001*; *Breaker and Joyce, 1994*; *Silverman, 2005*; *Chandra et al., 2009*). The oligomer replication based on catalyzed cleavage is not trivial as it requires cooperativity between multiple chain types. Our study shows that a stable cooperative solution can be achieved with as few as four subpopulations of chains. Furthermore, we demonstrate that there is a wide range of conditions under which this catalytic network proliferates and significantly outcompetes non-catalytic ancestors of the constituent chains.

## Acknowledgements

This research used resources of the Center for Functional Nanomaterials, which is a US DOE Office of Science User Facility, at Brookhaven National Laboratory under contract no. DE-SC0012704.

## Additional information

### Funding

| Funder | Grant reference number | Author |
| --- | --- | --- |
| Department of Energy Office of Science | DE-SC0012704 | Alexei V Tkachenko |

| Funder | Grant reference number | Author |
|---|---|---|

The funders had no role in study design, data collection and interpretation, or the decision to submit the work for publication.

## Author contributions

Alexei V Tkachenko, Sergei Maslov, Conceptualization, Software, Formal analysis, Investigation, Visualization, Writing – original draft, Writing – review and editing

## Author ORCIDs

Alexei V Tkachenko  http://orcid.org/0000-0003-1291-243X
Sergei Maslov  https://orcid.org/0000-0002-3701-492X

Reviewer #1 (Public Review): https://doi.org/10.7554/eLife.91397.3.sa1
Reviewer #3 (Public Review): https://doi.org/10.7554/eLife.91397.3.sa2
Author Response https://doi.org/10.7554/eLife.91397.3.sa3

# Additional files

## Supplementary files

• MDAR checklist

## Data availability

The current manuscript is a theoretical study, so no data have been generated for this manuscript.

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

# Appendix 1

## Effect of template rehybridization

Consider a simple model in which two mutually complementary sequences act as templates for each other's polymerization. Let $C$ and $\bar{C}$ be the concentrations of these unhybridized template chains. Let $c$ and $\bar{c}$ be the concentrations of unhybridized fragments of these two sequences, respectively. They may act as primers once bound to their complementary template. At the beginning of the 'night' phase, we assume all the chains to be in an unhybridized state, that is, $C = C_0$, $\bar{C} = \bar{C}_0$, $c = c_0$, and $\bar{c} = c_0$. We will describe the system's kinetics during the 'night' phase by assuming that hybridization is essentially irreversible, except for the possibility of a primer strain displacement by a full chain. We only focus on the regime when the total concentration of primers, $c_0$ and $\bar{c}_0$, is much smaller than that of free templates, $C$ and $\bar{C}$. Note that the concentrations of bound primers are $c_0 - c$ and $\bar{c}_0 - \bar{c}$, respectively.

The kinetics of this simple model is described by the following rate equations:

$$\dot{C} = -\kappa_0 C \bar{C} \tag{S1}$$

$$\dot{c} = -\kappa_1 \bar{C} c + \kappa_2 C (c_0 - c) \tag{S2}$$

The first equation accounts for the hybridization of mutually complementary template chains (assuming $c \ll C$ and $\bar{c} \ll \bar{C}$, $\kappa_0$ is the corresponding association rate). The second equation accounts for the hybridization of primers with their complementary substrates and for the primer strand displacement due to the hybridization of the template with its full-length complementary. The respective association rates for these two processes, $\kappa_1$ and $\kappa_2$, are dependent on the length and sequences of the chains involved, but generally comparable. The other two equations, for $\bar{C}$ and $\bar{c}$, are obtained by replacing all the concentrations with their complementaries.

We first consider the asymmetric case, when $C_0 > \bar{C}_0$ (without loss of generality). This implies that for a long enough duration of the night, the number of hybridized chains is limited by $\bar{C}_0$. Thus, only one type of the template chain will remain in the solution, $C = C_0 - \bar{C}_0$, while the other will be completely hybridized, that is, $\bar{C} = 0$. In turn, this implies that the situation will be even more drastic for the primers: $\bar{c} = 0$ (completely hybridized), while $c = c0$ (completely free). As a result, the minority fraction, $\bar{C}$ would replicate faster and the balance would be restored, so that $C_0 = \bar{C}_0$.

Now, we will analyze the above set of differential equations for a symmetric case, $C_0 = \bar{C}_0$. The fraction of free template chains vanishes with time as a power law (rather than exponentially):

$$C(t) = \bar{C}(t) = \frac{C_0}{1 + C_0 \kappa_0 t} \tag{S3}$$

As to the concentration of free fragments, it reaches a steady-state value between 0 and $c_0$:

$$c = \frac{c_0}{1 + \kappa_1/\kappa_2} \tag{S4}$$

Since the association rates $\kappa_1$ and $\kappa_2$ are comparable, we come to a surprising conclusion that a finite fraction of fragments, of the order of 1, will stay hybridized, despite the effect of strand displacement by longer chains:

$$\frac{c_0 - c}{c_0} = \frac{1}{1 + \kappa_2/\kappa_1} \sim 1 \tag{S5}$$

The major assumption behind this calculation is that primer concentration remains small compared to that of templates. Based on *Equation S3*, this leads to the following estimate of the optimal duration of the night phase in terms of primer concentration:

$$t_{night} \lesssim \frac{1}{\kappa c_0} \tag{S6}$$

By assuming $\kappa \sim 10^{-7} M^{-1} s^{-1}$, typical for RNA, and $t_{night} \sim 10^4 s$ (several hours), we conclude that the described mechanism is relevant for concentrations of specific primers, such as $a_R/a_L$ of the order of 10 pM. Note that (i) the concentration of templates (e.g., $a/\bar{a}$) would typically be significantly

larger, (ii) we only count chains with specific sequences, and (iii) this refers to the concentration of oligomers, not overall monomer concentration. This implies that the typical monomer concentration should be of nM scales or greater.

# Appendix 2

## Cooperative steady-state solution

The steady state of *Equations S7–S11* from the main text must satisfy the following set of equations:

$$\beta\phi a = \delta c - \beta_0 M \tag{S7}$$

$$\delta b = \beta_0 M\phi + \frac{rm\phi}{\lambda}a_R \tag{S8}$$

$$\beta\phi a = \left(rm + \delta\right) a_L \tag{S9}$$

$$\beta\phi a = \left(rm\left(1 - \phi + \frac{\phi}{\lambda}\right) + \delta\right) a_R \tag{S10}$$

$$rmc = \delta M \tag{S11}$$

Let us define

$$\mu = \frac{\delta}{rm}. \tag{S12}$$

By using *Equation S11* to exclude variable $M$, and replacing $b$ in *Equation S8* with $\phi(a + b)$, the fixed point conditions can be rewritten as

$$\beta\phi a = (\delta - \beta_0/\mu)c \tag{S13}$$

$$a + b = \frac{1}{\mu}\left(\frac{a_R}{\lambda} + \frac{\beta_0}{\delta}c\right) \tag{S14}$$

$$a_L = \frac{(\mu - \beta_0/\delta)c}{1 + \mu} \tag{S15}$$

$$a_R = \frac{(\mu - \beta_0/\delta)c}{1 + \mu - (1 - \lambda^{-1})\phi} \tag{S16}$$

As $a + b$, $a_L$, and $a_R$ are now expressed in terms of $c$, we use its definition, $c = a + b + a_L + a_R$, to get

$$\left(1 - \frac{\beta_0}{\mu\delta}\right)c = \left(1 + \frac{1}{\lambda\mu}\right)a_R + a_L = \left(1 - \frac{\beta_0}{\mu\delta}\right) \cdot$$
$$\cdot \left(\frac{1 + \lambda\mu}{1 + \mu - (\lambda - 1)\phi} + \frac{1}{1 + \mu}\right) \tag{S17}$$

This yields a compact expression for φ in terms of μ and $\lambda$ only, thus invariant with respect to both cleavage rates, $\beta_0$ and β:

$$\phi = \left(1 - \frac{\lambda\mu}{\lambda - 1}\right)(1 + \mu) \tag{S18}$$

Another relationship is obtained by using *Equation S14 and S16* to express $a = (1 - \phi)(a + b)$ in terms of $c$, and substituting it into *Equation S13*:

$$\frac{\delta}{\beta} = \phi(1 - \phi)\left(\frac{1}{\lambda\left(1 - \phi + \mu\right) + \phi} + \frac{1}{\mu\delta/\beta_0 - 1}\right) \tag{S19}$$

By substituting *Equation S18* into *Equation S19*, one gets the analytic relationship that allows to compute β for arbitrary values of μ, $\beta_0$, δ, and $\lambda$:

$$\frac{\delta}{\beta} = \frac{\mu}{\lambda - 1}\left(1 - \frac{\lambda\mu}{\lambda - 1}\right)\left(1 + (1 + \mu)(1 + \lambda\mu)\frac{\beta_0}{\mu\delta - \beta_0}\right) \tag{S20}$$

In practice, it is the monomer concentration that gets adjusted to its steady-state value $m^*$, thus μ is the variable that has to satisfy the above equation, for given values of other parameters.

The stable fixed points only appear on the decreasing segment of the function $\beta(\mu)$. Thus, the limits of stability correspond to zero derivatives of the r.h.s. of **Equation S20**. This leads to the following condition:

$$\left(1 - \frac{2\lambda\mu}{\lambda - 1}\right)z^2 + \mu A'(\mu)z - A(\mu) = 0 \tag{S21}$$

Here we defined $z = \mu\delta/\beta_0 - 1$,

$$A(\mu) = \left(1 - \frac{\lambda\mu}{\lambda - 1}\right)(1 + \mu)(1 + \lambda\mu)$$
$$= 1 + \frac{(\lambda^2 - \lambda - 1)\mu - 2\lambda\mu^2 - \lambda^2\mu^3}{\lambda - 1} \tag{S22}$$

$$A'(\mu) = \frac{(\lambda^2 - \lambda - 1) - 4\lambda\mu - 3\lambda^2\mu^2}{\lambda - 1} \tag{S23}$$

The positive solution to the quadratic **Equation S21** is given by

$$z(\mu) = \frac{(\lambda - 1)\mu A'(\mu)}{2(\lambda - 1 - 2\lambda\mu)}$$
$$\cdot\left(\sqrt{1 + 4\left(1 - \frac{2\lambda\mu}{\lambda - 1}\right)\frac{A(\mu)}{\mu^2 A'(\mu)^2}} - 1\right) \tag{S24}$$

Now, by using **Equation S20** and our definition of $x$, both β and $\beta_0$ can be parameterized by μ:

$$\beta_0(\mu) = \frac{\mu\delta}{z(\mu) + 1} \tag{S25}$$

$$\beta(\mu) = \frac{\delta(\lambda - 1)}{\mu\left(1 - \frac{\lambda\mu}{\lambda - 1} + \frac{A(\mu)}{z(\mu)}\right)} \tag{S26}$$

This parametric curve in $(\beta, \beta_0)$ space defines the boundary of the region in which the non-trivial fixed point solution exists.

A simplified asymptotic relationship between β, $\beta_0$, and μ can be obtained in the limit of $\mu \ll 1$. In this case, **Equation S24** implies $z(\mu) \approx 1$. By substituting this result into **Equations S25 and S26**, one obtains

$$\mu \approx 2\beta_0/\delta \tag{S27}$$

$$\beta \approx \frac{(\lambda - 1)\delta^2}{4\beta_0} \tag{S28}$$

This result gives the lower bounds for μ and the upper bound for β consistent with the cooperative solution, for a given $\beta_0$. The approximation is valid in the limit of $\mu \ll 1$, that is, $\beta_0/\delta \ll 1$.

Another asymptotic result can be obtained for vanishingly small non-catalytic cleavage rate $\beta_0$. In that limit, **Equation S20** turns into quadratic equation since $(1 + \mu)(1 + \lambda\mu)/\mu \ll \delta/\beta_0$. Its two solutions correspond to unstable and stable fixed points. Specifically, the stable branch is given by

$$\mu \approx \frac{\lambda - 1}{2\lambda}\left(1 - \sqrt{1 - \frac{4\lambda\delta}{\beta}}\right) \tag{S29}$$

The solution only exists for $\beta/\delta \geq 4\lambda$, which sets the lower bound for β. This critical point corresponds to $\mu = \frac{\lambda - 1}{2\lambda}$, which is the upper bound of μ. Note that this approximation is valid as long as

$$\frac{\beta_0}{\delta} \ll \frac{\mu}{(1+\mu)(1+\lambda\mu)} \approx \frac{2(\lambda-1)}{(3\lambda-1)(\lambda+1)} \tag{S30}$$

To summarize, the range of β for which the cooperative solution exists, and the respective range of the steady-state monomer concentration $m_{\text{coop}}^*$, can be approximated as

$$4\lambda < \frac{\beta}{\delta} < \frac{(\lambda-1)\delta}{4\beta_0} \tag{S31}$$

$$\frac{2\lambda\delta}{(\lambda-1)\beta_0} < \frac{m_{\text{coop}}^*}{m^*} < 1/2 \tag{S32}$$

Here, $m^* = \delta^2/(r\beta_0)$ is the steady-state monomer concentration in the non-cooperative regime given by *Equation 5*.

The fitness parameter for the cooperative regime

$$\frac{m^*}{m_{\text{coop}}^*} = \frac{\mu\delta}{\beta_0} \tag{S33}$$

