## [Editor Report · eLife assessment]

This **valuable** study uses a model to determine when catalytic self-replication of polymers can emerge from a random pool of replicating polymers. The model accounts for the folding and function of polymers in addition to abstract evolutionary dynamics, providing **solid** evidence for the claims of the authors. The work will be of relevance to those interested in the origin of life, artificial cells, and evolutionary dynamics.

---

## [Referee Report · Reviewer #1 (Public Review)]

Summary:

The emergence of catalytic self-replication of polymers is an important question in the context of the origin of life. Tkachenko and Maslov present a model in which such a catalytic polymer sequence emerges from a random pool of replicating polymers.

Strengths:

The model is part of a theme from many previous papers from the same authors and their colleagues. The model is interesting, technically correct and demonstrates qualitatively new phenomena. It is good that the paper also makes a connection with possible experimental scenarios - specifically, concrete proposals are made for testing the core ideas of the model. It would indeed be an exciting demonstration when such an experiment does indeed materialize.

Weaknesses:

Unlike the rest of the paper which is very tight in its arguments, I find that the discussion section is not so. Specifically, sentences such as " In fact, this can be seen as a special case of the classical error catastrophe" are a bit loose and not well substantiated -- although these are in the discussion section, I find this to be a weakness of an otherwise good paper and tightening some of the arguments here will make it an excellent paper in my opinion.

---

## [Referee Report · Reviewer #3 (Public Review)]

Summary:

Non enzymatic replication of RNA or a similar polymer is likely to be important for the origin of life. The authors present a model of how a functional catalytic sequence could emerge from a mixture of sequences undergoing non-enzymatic replication.

Strengths:

Interesting model describing details of the proposed replication mechanism.

Weaknesses:

The idea of the virtual circular genome proposed in [37] is included in the discussion section together with the problem of sequence scrambling faced by this mechanism that was raised in [38]. Sequence scrambling arises in models that assume cycles of melting and reannealing, in which case only part of a template is copied in one cycle. Scrambling is due to the many alternative ways in which pairs of sequences can reanneal. Many of these alternatives are incorrect and this leads to the disappearance of the original sequence. This problem exists even in the limit where there is zero mutational error rate. Thus, it is a separate problem from the usual error threshold problem. Scrambling would not occur if there was complete copying of a template from one end to the other.

The authors seem to believe that their model avoids the scrambling problem to some extent. If I understand correctly, this is because the functional activity is located in a short sequence region. I can imagine that if the length of a strand that is synthesized in a single melting/annealing cycle is long enough to cover the complete functional region, then sometimes the complete functional sequence can be copied in one cycle. The authors give an estimate of a scrambling-free length. I am not sure how this is determined. I think that the problem of how to encode functional sequences in RNA strands undergoing non-enzymatic replication is still not fully resolved.

---

## [Author Response]

The following is the authors’ response to the original reviews.

**Public Reviews:**

**Reviewer #1 (Public Review):**
Summary:The emergence of catalytic self-replication of polymers is an important question in the context of the origin of life. Tkachenko and Maslov present a model in which such a catalytic polymer sequence emerges from a random pool of replicating polymers.Strengths:The model is part of a theme from many previous papers from the same authors and their colleagues. The model is interesting, technically correct, and demonstrates qualitatively new phenomena. It is good that the paper also makes a connection with possible experimental scenarios -- specifically, concrete proposals are made for testing the core ideas of the model. It would indeed be an exciting demonstration when such an experiment does indeed materialize.Weaknesses:Unlike the rest of the paper which is very tight in its arguments, I find that the discussion section is not so. Specifically, sentences such as " In fact, this can be seen as a special case of the classical error catastrophe" are a bit loose and not well substantiated -- although these are in the discussion section, I find this to be a weakness of an otherwise good paper. Tightening some of the arguments here will make it an excellent paper in my opinion.

We followed the reviewer's recommendations by streamlining the discussion and removing the potentially confusing comparison to the classic error catastrophe.

**Reviewer #2 (Public Review):**
Summary:The replication of information-coding polymers and the emergence of catalytic ribozymes pose significant challenges, both experimentally and theoretically, in the study of the RNA world hypothesis. In this context, Tkachenko et al. put forth a novel hypothesis regarding a replication oligomer system based on a cleavage ribozyme. They initially highlighted that the breakage of oligomers could contribute to self-replication, provided that these fragments function as primers for subsequent replications. Next, they proposed a self-replicating system of oligomers founded on a hammerhead structure that catalyzes cleavage. By a simple dynamical model, they demonstrated that such a system is self-sustainable in certain parameter regimes. Furthermore, they delved into discussions regarding the potential emergence of such a system and the evolution toward further optimized ribozymes.Strengths:Although the cleavage (hammerhead) ribozyme has been discussed in the context of the origins of life, the authors are the first to discuss how they could be selected using a mathematical model as far as I know. The idea is simple: ribozyme activity creates fragments by breakage of an oligomer, which works as a primer for the ribozyme itself, resulting in a positive feedback system (i.e., autocatalytic sets in a broader sense). This potentially enables us to resolve at the same time problems on the (i) supply of new primers (but note that there is a major concern on this as described in the 'weakness'), and (ii) the sustaining of the cleavage ribozyme.Weaknesses:The major weakness of their theory is that the ends of the new primers, formed through the breakage/cleavage of polymers, must be chemically active (as the authors have already emphasized in the last paragraph of their discussion) to enable further elongation. Reactivating the ends of preexisting oligomers without enzymes, to the best of our current knowledge, could be a challenging task. Although their model heavily relies on this aspect, the authors do not elaborate on it.

We have added a discussion of the need for chemical activation: "It is important to note that in the context of RNA, such bidirectional elongation requires chemical activation of the phosphate group at the 5' end of the primer to provide free energy for the newly formed covalent bond. Like the polymerization process itself, achieving this without enzymes is biochemically challenging. One might speculate that prebiotic evolution relied on inorganic catalysis, such as on mineral surfaces, or involved polymers other than today's RNA."

We also included in the discussion a comment on a possible combination of our mechanism and the Virtual Circle Genome model that would avoid the need for bidirectional growth: "It may be possible to incorporate the selection mechanism proposed in this paper into the Virtual Circle Genome model. Such a hybrid approach would avoid the need for the biochemically problematic bidirectional growth while explaining the emergence of early catalytic activity unaffected by sequence scrambling"

Another weakness is in the setup of their discussion on evolutionary dynamics. While they claim that their model is robust against replication errors, their approach to evolutionary dynamics appears unconventional, and it remains unclear under what conditions their assumptions are founded. They treat a whole set of oligos as a subject of evolution, rather than each individual oligo. This may necessitate more complex assumptions, such as the encapsulation of sets of oligos inside a protocell, to be adequately rationalized. Thus, it remains uncertain whether the system is indeed robust against replication errors in a more natural context. For example, if a mutant oligo, denoted as b', arises due to an error in the replication of oligo b, and if b' has lower catalytic activity but replicates more rapidly than b, it may ultimately come to dominate the system.

We agree with the reviewer that the evolutionary dynamics in multi-species ecosystems are somewhat complicated and potentially confusing. To this end, we have added the following text and citations to our discussion: "Note that this fitness is defined at the level of the ecosystem, comprising all sequences in the chemostat, and is not necessarily attributable to individual members of that population. Over time, similar to microbial ecosystems, this population changes according to the laws of competitive exclusion [34, 35]". However, we would like to point out that we assume that our model operates in a chemostat-like environment, which can be realized, for example, in a prebiotic pool supplied with a constant flux of monomers. Thus, the evolutionary dynamics described by our equations do not require encapsulation of sets of oligos in a protocell followed by selection of these protocells.

**Reviewer #3 (Public Review):**
Summary:Non-enzymatic replication of RNA or a similar polymer is likely to be important for the origin of life. The authors present a model of how a functional catalytic sequence could emerge from amixture of sequences undergoing non-enzymatic replication.Strengths:Interesting model describing details of the proposed replication mechanism.Weaknesses:A discussion of the virtual circular genome idea proposed in [33] is included in the discussion section together with the problem of sequence scrambling faced by this mechanism that was raised in [34]. However, the authors state that sequence scrambling is a special case of the classical error catastrophe. This should be reworded, because these phenomena are completely different. The error catastrophe occurs due to single-point mutational errors in a model that assumes that a complete template is being copied in one cycle. Sequence scrambling arises in models that assume cycles of melting and reannealing, in which case only part of a template is copied in one cycle. Scrambling is due to the many alternative ways in which pairs of sequences can reanneal. Many of these alternatives are incorrect and this leads to the disappearance of the original sequence. This problem exists even in the limit where there is zero mutational error rate. Therefore, it cannot be called a special case of the error catastrophe problem.

We followed the reviewer's recommendations and removed the potentially confusing comparison to the classic error catastrophe.

The authors seem to believe that their model avoids the scrambling problem. If this is the case, a clear explanation should be added about why this problem is avoided. Two possible points are mentioned.(i) Replication is bidirectional in this model. This seems like a small detail to me. I don't think it makes any difference to whether scrambling occurs.(ii) The functional activity is located in a short sequence region. I can imagine that if the length of a strand that is synthesized in a single cycle is long enough to cover the complete functional region, then sometimes the complete functional sequence can be copied in one cycle. Is this what is being argued? If so, it depends a lot on rates of primer extension and lengths of melting cycles etc, and some comment on this should be made.

As we now explain in the text, while the scrambling problem itself is not completely avoided in our model, it does not affect the replication of the functionally relevant regions of the oligomers. Our key observation is that, due to the simplicity of the cleaving enzymes, the length of the functionally relevant region is much smaller than the scrambling-free length. This can be seen from a back-of-the-envelope estimate of the scrambling-free length added to the text: "...assuming the minimal hybridization length l0=6 and random statistics of the master sequence, one gets the scrambling free length 2x40l+l0 100. This is an order of magnitude larger than both l0 and the length of the core region of the hammerhead ribozyme."

**Recommendations for the authors:**

**Reviewer #2 (Recommendations For The Authors):**
I have evaluated that the authors have proposed a novel mechanism potentially relevant to the origins of life, and they have explained it with a sufficiently simple model. However, I recommend that they address the following issues, including those I raised in the public review:Title: I believe that the title "Emergence of catalytic activity in ..." is rather broad. Could it be more specific to accurately represent the system described in the paper? For instance, "Selective advantage (or selection) of the hammerhead cleavage ribozyme in..." may better encapsulate the paper's focus.

We thank the reviewer for this suggestion. However, our mechanism is not unique to hammerhead ribozymes. So we decided to keep the old title.

One theoretically non-trivial aspect is the stability of the cooperative structure. Could the authors provide a more detailed explanation of what drives the instability of the system and what mechanisms restore its stability? For example, in a similar self-reproducing oligomer system with ribozymes and their fragments (Kamimura et al. PLoS Comp. 2019), the symmetry of fragments breaks because they effectively suppress each other's replication. Also, it would be beneficial to clarify the necessary assumptions for stability. (For instance, the authors assumed that a_L can serve as a primer for only a, while a_R can serve for both a and b.).

We thank the reviewer for bringing this interesting paper to our attention. The cooperative fixed point in our model is intrinsically dynamically stable. It is an interesting point why the replicase in Kamimura et al can be dynamically unstable, while the ligase in our model is always stable. However, it goes beyond the scope of our study. We added the following discussion to the manuscript: "Note that the stability of our cooperative fixed point is a non-trivial result. For example, in a related model by Kamimura et al. [34], the fixed point corresponding to a viable composite replicase is dynamically unstable and requires additional stabilization, e.g., by cell-like compartments."

As mentioned in the public review, a critical aspect of the practical applicability of the theory is whether cleaved oligos can be reactivated and further elongated, especially through non-enzymatic pathways. Alternatively, is it possible with the presence of enzymes? While I appreciate the conceptual beauty of their model, I recommend that they at least address the difficulty or feasibility of achieving this.

We addressed this point in response to the public review

As also mentioned, in the section on evolutionary dynamics, it's essential to clarify the unit of evolution and the assumptions made. For a system-level evolution (i.e., all the sets of oligos, a and b can be the unit of evolution), more detailed assumptions are required, such as the presence of compartments whose growth is coupled with the replication of oligos inside, and the competition between these compartments. I recommend the authors clarify these points.

We addressed this point in response to the public review

**Reviewer #3 (Recommendations For The Authors):**
Assuming that the above points can be addressed, this reviewer would support publication with minor modifications.

We addressed all points in response to the public review